# Thermoelectric Properties of Cotton Fabrics Dip-Coated in Pyrolytically Stripped Pyrograf^®^ III Carbon Nanofiber Based Aqueous Inks

**DOI:** 10.3390/ma16124335

**Published:** 2023-06-12

**Authors:** Antonio J. Paleo, Beate Krause, Maria F. Cerqueira, Jose M. González-Domínguez, Enrique Muñoz, Petra Pötschke, Ana M. Rocha

**Affiliations:** 12C2T-Centre for Textile Science and Technology, University of Minho, Campus de Azurém, 4800-058 Guimarães, Portugal; amrocha@det.uminho.pt; 2Leibniz-Institut für Polymerforschung Dresden e.V. (IPF), Hohe Str. 6, 01069 Dresden, Germany; krause-beate@ipfdd.de (B.K.); poe@ipfdd.de (P.P.); 3INL-International Iberian Nanotechnology Laboratory, Av. Mestre. Jose Veiga, 4715-330 Braga, Portugal; fcerqueira@fisica.uminho.pt; 4CFUM–Center of Physics of the University of Minho, Campus de Gualtar, 4710-057 Braga, Portugal; 5Instituto de Carboquímica ICB-CSIC, C/Miguel Luesma Castán 4, 50018 Zaragoza, Spain; jmgonzalez@icb.csic.es; 6Facultad de Física, Pontificia Universidad Católica de Chile, Santiago 7820436, Chile; munoztavera@gmail.com

**Keywords:** carbon nanofibers, dip-coating, conductive textiles, Seebeck coefficient, variable range hopping model

## Abstract

The transport properties of commercial carbon nanofibers (CNFs) produced by chemical vapor deposition (CVD) depend on the various conditions used during their growth and post-growth synthesis, which also affect their derivate CNF-based textile fabrics. Here, the production and thermoelectric (TE) properties of cotton woven fabrics (CWFs) functionalized with aqueous inks made from different amounts of pyrolytically stripped (PS) Pyrograf^®^ III PR 25 PS XT CNFs via dip-coating method are presented. At 30 °C and depending on the CNF content used in the dispersions, the modified textiles show electrical conductivities (σ) varying between ~5 and 23 S m^−1^ with a constant negative Seebeck coefficient (S) of −1.1 μVK^−1^. Moreover, unlike the as-received CNFs, the functionalized textiles present an increase in their σ from 30 °C to 100 °C (dσ/dT > 0), explained by the 3D variable range hopping (VRH) model as the charge carriers going beyond an aleatory network of potential wells by thermally activated hopping. However, as it happens with the CNFs, the dip-coated textiles show an increment in their S with temperature (dS/dT > 0) successfully fitted with the model proposed for some doped multiwall carbon nanotube (MWCNT) mats. All these results are presented with the aim of discerning the authentic function of this type of pyrolytically stripped Pyrograf^®^ III CNFs on the thermoelectric properties of their derived textiles.

## 1. Introduction

The addition of the electrical conductivity functionality to the intrinsic properties of textiles (i.e., flexibility, strength, comfort, and wearability, etc.) has attracted special attention due to the great potential for applications as diverse as sensors, actuators, antennas, batteries, etc. [1,2]. Overall, their production requires conductive materials and the application of specific processes to embed them in textiles. With respect to the materials with electrical conductivity, they can be divided into two main categories: organic materials such as conjugated polymers and inorganic materials such as metals, metal oxides, transition metal dichalcogenides (TMDs), etc. [3]. In addition, carbon materials such as carbon black (CB), graphite, carbon fibers (CFs), etc., as well as nanomaterials such as carbon nanofibers (CNFs), carbon nanotubes (CNTs), graphene, etc. are largely used due to their affordability and outstanding electrical properties [4,5]. Unfortunately, most of them are in solid form, which makes their integration into textiles challenging. Thereby, one common strategy is the production of inks composed of the conductive materials together with resins and oils used as binders and/or carriers [6,7]. As such, the particular ink can be integrated in the textile through methods as diverse as screen-printing, inkjet-printing, spray-coating, and dip-coating [2,8]. More importantly, the relationships between the electronic properties of the conductive materials and the resulting conductive textiles need to be well understood in order to make convincing progress for a given target application. In this respect, it is noted that, among the numerous applications in which conductive textiles can be utilized, the focus has recently been directed towards thermoelectric (TE) textiles capable of converting body heat into electricity by exploiting the Seebeck effect [9,10]. This Seebeck effect is quantified by the Seebeck coefficient (S), which is the quotient of the thermoelectric voltage (ΔV) generated by the TE textile at a temperature difference (ΔT) applied between its ends. It should be noted that the sign of S can be used to differentiate whether a TE textile is n-type or p-type. Hence, in n-type TE textiles (negative S), the majority of the charge carriers are electrons (e^−^), while in p-type TE textiles (positive S), holes are the dominant charge carriers [11,12]. Additionally, the efficiency of a TE textile is determined by its dimensionless figure of merit (zT), defined as $zT=\frac{S^{2}\sigma }{k}T$, where σ is the electrical conductivity, k is the thermal conductivity, and T is the absolute temperature [13]. This leads to the conclusion that, in order to obtain textiles with high TE functionality, a high S and σ together with a low k are imperative. The present study therefore situates within this framework, and aligns with previously reported studies [14,15], in which the Pyrograf^®^ III PR 24 LHT XT CNF grade was applied to produce electrically conductive textiles based on cotton woven fabrics (CWFs) by dip-coating. Now, aqueous inks made by dispersing three different contents of pyrolytically stripped (PS) Pyrograf^®^ III PR 25 PS XT CNFs (another grade of Pyrograf^®^ III CNFs) with sodium dodecylbenzenesulfonate (SDBS) are used to prepare CWFs with electrical functionality. Then, a comparison between σ and S from 30 °C to 100 °C of the as-received CNFs and the electrically conductive textiles prepared with the aqueous CNF inks is discussed. In addition, the 3D variable range hopping (VRH) [16] and the model describing the nonlinear Seebeck of doped MWCNT mats [17] are applied to CNFs and derivative conductive textiles to derive insights into the underlying physics of their σ (T) and S (T). The ultimate goal is to derive insights to the extent to which the final TE properties of the resulting conductive coated textiles depend on the type of Pyrograf^®^ III CNFs used in their production.

## 2. Materials and Methods

### 2.1. Materials

The 100% cotton woven fabric (CWF) (Somelos-Tecidos, S.A, Ronfe, Portugal) was used as-received. Its physical characteristics have been reported in previous work [15]. The CWF has weft and warp yarns with air-filled spaces of about 250 × 250 μm^2^ located among them (Figure 1a). The carbon nanofibers used in this work, which are labeled as Pyrograf^®^ III, are manufactured by Applied Sciences, Inc. (Cedarville, OH, USA) and differ in their diameter (PR 24 < PR 19~PR 25) and grade (PS, LHT, or HHT) [18]. Among them, the PR 25 PS XT type was selected to provide the CWF with electrical properties. Specifically, these CNFs are grown by CVD at 1100 °C, and subjected to a thermal post-treatment of 700 °C in an inert atmosphere. The term PS (of PR 25 PS XT) indicates that they have been subjected to pyrolytic stripping to eliminate the polyaromatic hydrocarbons from the surface, whereas the term XT refers to the debulked form of the PR 25 family [19]. These CNFs show a double-wall structure surrounding the hollow core (Figure 1b). Some of their properties such as bulk density (0.0192–0.0480 g cm^−3^); density (including hollow core) from 1.4 to 1.6 g cm^−3^; outer diameter from 125 to 150 nm; average inner diameter from 50 to 70 nm; and lengths ranging from 50 to 100 μm have been listed elsewhere [19]. The other materials used were obtained from Sigma-Aldrich without further purification.

### 2.2. Preparation of Conductive Textiles

The dip-coating method used for the functionalization of the textiles was the same as that reported in previous works [14,15]. Three dispersions in distilled water (DI) with 1.6, 3.2, and 6.4 mg·mL^−1^ of CNFs and 5 mg·mL^−1^ of sodium dodecylbenzenesulfonate (SDBS), were produced in a first step involving tip sonication (ultrasonic homogenizer CY-500; 60% power, 5 min). Immediately after, six pieces of CWF samples (2 × 2 cm^2^) were dipped in that ink for 5 min, and then dried at 80 °C for 10 min. This dip-coating step was made five times to ensure that the CWF properly absorbs the CNFs. The samples are subjected four times to washing, consisting of dipping during 10 min in DI, followed by a drying at 80 °C for 10 min. Finally, dipping in ethanol and drying at 80 °C during 10 min was performed to warrant the removal of SDBS. At the end, three types of dip-coated cotton fabrics, henceforth referred to as CWF@1.6 CNF, CWF@3.2 CNF, and CWF@6.4 CNF, are utilized for the TE analysis.

### 2.3. Morphological and Structural Analysis

The as-received CNFs are observed with a transmission electron microscope (TEM) (JEM-2100, JEOL Ltd., Tokyo, Japan) operating a LaB6 electron gun at 80 kV. Images were acquired with “OneView” 4 k × 4 k CCD camera at a minimal under-focus to achieve the visibility of the CNF surface layers. The morphological analysis of CWFs was carried out in an ultra-high-resolution field emission gun scanning electron microscopy (FEG-SEM) (NOVA 200 Nano SEM, FEI Company, Hillsboro, OR, USA). Pieces of the dip-coated textiles were placed flat on an SEM grid before examining them using field emission scanning electron microscopy (SEM) (microscope Ultra plus, Zeiss, Oberkochen, Germany) at 3 kV in combination with SE2 detector. The Raman spectroscopy measurements (ALPHA300 R Confocal Raman Microscope, WITec GmbH, Ulm, Germany) of CNFs and dip-coated textiles were carried out at room temperature in a back-scattering geometry using a 532 nm Nd:YAG laser for excitation. The laser beam with *P* = 0.5 mW was focused on the sample by a ×50 lens (Zeiss), and the spectra were collected with 600 groove/mm grating using 5 acquisitions with 2 s acquisition time. X-ray photoelectron spectroscopy (XPS) measurements were performed in an ultra-high vacuum (UHV) system (ESCALAB250Xi, Thermo Fisher Scientific, Waltham, MA EUA). The base pressure in the system was below 5 × 10^−10^ mbar. XPS spectra were acquired with a hemispherical analyzer and a monochromated X-ray source (Al K_α_ radiation, hν = 1486.6 eV) operated at 15 keV and a power of 200 W.

### 2.4. Thermoelectric Analysis

The Seebeck coefficient (S) and electrical volume resistivity (ρ) of the CNF powder and the functionalized textiles were determined using the self-constructed equipment TEG developed at Leibniz-IPF [20,21]. The values of volume resistivity analyzed by the four-wire technique represent the arithmetic mean values of 10 measurements. This procedure was repeated 5 times, and the mean values and standard deviation were then calculated. The values of Seebeck coefficient were made on a sample piece and repeated 5 times. Modified textiles with a size of ca. 22 mm width cut from the dipped fabrics using scissors were inserted between two copper electrodes, which have a distance of about 18 mm between them. For the CNF powder, the experiments were performed using an insert consisting of a PVDF tube (inner diameter of 3.8 mm, length of 16 mm) filled with the CNF powder and closed with copper plugs, which were also inserted between the two copper electrodes [22]. The S is measured by implementing temperature gradients between the two copper electrodes of up to ±8 K (eight steps of 2 K each around the mean temperatures from 30 °C to 100 °C), and is then calculated as the average of 10 thermovoltage measurements. The figure of merit is estimated using a through-plane thermal conductivity of 0.43 W m^−1^ K^−1^, which was obtained for anisotropic paper-like mats of Pyrograf^®^ III PR-25 CNFs [23].

## 3. Results and Discussion

### 3.1. Morphological Analysis of As-Received CNFs and Dip-Coated Textiles

The as-received CNFs used for the production of the aqueous inks are observed with TEM as described in Section 2.3. A hollow core surrounded by two layers is observed in the single CNF (Figure 1b). The internal layer with a thickness of ~15 nm shows a group of highly compacted parallel graphene plies with a determined angle with respect to the hollow core axis, contrary to the outer wall layer, which presents a lower thickness of ~9 nm, and is clearly more disordered. Overall, the total diameter was around 120 nm, and therefore, compared to other grades of Pyrograf^®^ III, such as PR 24 and PR 19, the PR 25 grade has a diameter comparable to PR 19, but is higher than PR 24 [18]. Figure 2 shows the SEM images of dip-coated textiles used for the thermoelectric analysis. The original morphology of the CWF (Figure 1a) can be clearly noticed on the surface of CWF@1.6 CNF (Figure 2a), whereas a kind of CNF mat partially and completely covers the original CWF surface in CWF@3.2 CNF (Figure 2b) and CWF@6.4 CNF (Figure 2c), respectively. Notably, the mats visible in Figure 2b,c are not continuous, but appear as a kind of intercalated CNF flakes. This is a clear sign that CNFs are not perfectly bonded to the CWF. The functional ink prepared with the highest content of CNFs (6.4 mg∙mL^−1^) should a priori yield the textiles with the highest amount of CNFs entrapped in the 3D structure of the CWFs. This is confirmed by measuring their thicknesses. It is recalled that the uncoated CWF has a thickness of 0.26 mm [14,15]. It is observed that the thicknesses of CWF@1.6 CNF and CWF@3.2 CNF do not differ greatly, with values of 0.37 ± 0.02 and 0.38 ± 0.01 mm, respectively, and the CWF@6.4 CNF has the largest thickness of 0.43 ± 0.01 mm. Interestingly, a comparable thickness of 0.47 ± 0.04 mm was measured for CWF@6.4 CNF samples obtained by the same dip-coating process and using the same CWF, but with a different type of Pyrograf^®^ III CNFs (PR 24 LHT XT) [15]. In summary, the different contents of CNFs used for the preparation of the functional aqueous inks clearly influence the surface morphology of the electrically conductive textiles. Moreover, from the SEM images and the thicknesses before and after dip-coating, it can be deduced that the CWF@6.4 CNF sample retains a larger amount of CNFs. It can also be concluded that the CNFs do not strongly adhere to the CWF for the CWF@3.2 CNF and CWF@6.4 CNF samples, which may affect their electric properties when subjected to bending loads.

### 3.2. Raman Analysis of As-Received CNFs and Dip-Coated Textiles

Figure 3 shows the Raman spectra in the range between 200 and 3000 cm^−1^ of the as-received CNFs used for the production of the aqueous inks and dip-coated textiles used for the thermoelectric analysis. The Raman spectra of carbon-based materials is characterized by the presence of two main bands, the G-band (E_2g_ symmetry) at 1580 cm^−1^, characteristic of the ideal graphitic lattice vibration [24], and the D-band (A_1g_ symmetry) around 1350 cm^−1^, detected when structural defects are present in the carbon aromatic structure [25]. In particular, the relative motion of the sp^2^-bonded carbon atoms causes the G–band, while the D-band is related to the presence of six-membered rings [26]. The Raman spectra of CWF (Figure 1a) has been already reported and it shows the characteristic modes observed in cellulose [14,15]. As expected, all samples (CNFs and functionalized textiles) show the G and D modes, as well as the presence of the 2D mode. Naturally, the presence of cellulose from the CWF is not detected in the coated textiles, since the Raman resonance in CNFs is much more intense than in cellulose. The 2D mode (second-order Raman mode) is also characteristic of carbon-based materials, and it is associated with the more intense mode of monolayer graphene [27]. The peak position and the full width at half maximum (FWHM) of the modes were determined by fitting the Raman spectra with Lorentzian functions (see Appendix A). The derived parameters are shown in Table 1 together with the in-plane graphitic domain size (L_a_) determined by L_a_ (nm) = 4.4/(I_D_/I_G_) [28]. It is noteworthy that the FWHM_G_ and FWHM_D_ are approximately the same in all of the samples, whereas the FWHM_D_ is higher than the FWHM_G_. The intensity ratios between the D and G bands (I_D_/I_G_) are also presented in Table 1 since they are relevant parameters to quantify the number of disordered (D) and ordered (G) carbon atoms [29]. As expected, the I_D_/I_G_ of this CNF is higher than the I_D_/I_G_ of the CNF grade used in the previous works [14,15], associated with the lower levels of graphitization of this grade (PR 25 PS XT). Moreover, the results show a decrease in the I_D_/I_G_ ratio and an increase in L_a_ for the coated textiles when compared to the CNFs. Both findings are a signal of a higher regularity in the carbon network, which can be provoked by the structure of CWF [14]. As a principal conclusion, the three functionalized textiles share practically the same Raman spectra with no significant changes caused by the lower or higher amount of CNFs used in the preparation of the inks.

### 3.3. XPS Analysis of As-Received CNFs and Dip-Coated Textiles

Figure 4 shows the analysis by X-ray photoelectron spectroscopy of the as-received CNFs and CWF@1.6 CNF sample. XPS is a characterization technique useful to ascertain the chemical nature of surfaces in (nano)materials, and in this case, it could hint at the successful achievement of hybrid CWF@CNF fabrics. The pristine CNFs show a high carbon content (96 at %), with around 3% oxygen according to the survey spectrum (see Appendix A). This would agree well with a high-purity carbonaceous material, in which the majority of carbon atoms are in sp^2^ hybridization, C=C bonds, as seen at 284.5 eV (Figure 4a). The low oxygen functionality of CNFs can be ascribed to alcohols/phenols (as judged by the contribution at ~286 eV), and COO groups, as seen at 290.3 (Figure 4a), 531.7, and 533.3 eV (Figure 4c) [30]. However, it is noticed that the CNF grade used in the previous works (PR 24 LHT XT) presents even an lower oxygen content (1.8%) [14,15]. As an example of the CWF@CNF samples, Figure 4b,d show the C and O core levels of CWF@1.6 CNF, respectively. The profile still retains some of the features from the CNFs and displays a higher oxygen content (C/O ratio = 78/16) according to the survey spectrum (see Appendix A). Indeed, the chemical nature of this sample resembles the native cotton fabric (see its XPS spectra elsewhere [15]), although the CWF@1.6 CNF samples prepared with the PR 24 LHT XT grade [15] present slightly lower oxygen content (C/O ratio = 67/13). The C=C component at 284.5 eV is still dominant, while the nature of oxygen groups varies towards a slight amount of carboxylic groups at 534.9 eV and a large amount of –OH groups at 532.6 eV (Figure 4d) [30]. The oxygen groups’ nature is compatible with the composition of the native cotton fabric [15]. Thus, in general terms, the CWF@CNF hybrid textile shows combined features according to both constituent materials, and it could point to a reasonably well-integrated system.

### 3.4. Thermoelectric Properties of As-Received CNFs and Dip-Coated Textiles at 30 °C

#### 3.4.1. Electrical Conductivity

Figure 5 and Table 2 present the experimental results corresponding to the thermoelectric properties at 30 °C of the coated textiles and the as-received CNF powder. The electrical conductivity of the CNF powder is 136.1 ± 12 S∙m^−1^ (~7 × 10^−1^ Ohm∙cm), while the single Pyrograf^®^ III CNFs are reported to achieve values of 4 × 10^−3^ Ohm cm [31]. This lower σ with respect to individual CNFs can be related to the CNF powder measured in this study, which represents a packing of CNF agglomerates. The measured CNF powder conductivity is similar to the σ of the other Pyrograf^®^ III CNF grade (PR 24 LHT XT) used in the previous work [15], and lower than the electrical powder conductivity of ~417 S∙m^−1^ obtained for some commercial multiwall carbon nanotubes (MWCNTs) such as NC7000^TM^ [22].

The functionalized textiles have σ from 5.14 ± 0.29 S∙m^−1^ of CWF@1.6 CNF to 23.19 ± 0.68 S∙m^−1^ of CWF@6.4 CNF samples (Table 2). Thus, their σ is significantly lower than the σ of the CNF powder used for the preparation of the inks. This is not surprising, as the non-conductive character of the CWF must hinder the appropriate creation of electronic pathways between CNFs. Clearly, the higher content of CNFs (6.4 mg∙mL^−1^) utilized in the inks of CWF@6.4 CNF explains their higher σ (when compared to CWF@1.6 CNF and CWF@3.2 CNF samples). Nevertheless, the conductivity achieved for CWF@6.4 CNF (23.2 S∙m^−1^) is lower than the values of 35.4 S∙m^−1^ obtained for samples prepared in the same manner with 6.4 mg∙mL^−1^ of Pyrograf^®^ III PR 24 LHT XT CNFs [15].

#### 3.4.2. Seebeck Coefficient

Figure 5 and Table 2 show the experimental Seebeck coefficient of the coated textiles and as-received CNF powder at 30 °C. The CNF powder shows n-type character (−0.60 ± 0.03 μV∙K^−1^), which is in agreement with other Pyrograf^®^ III CNF grades (PR 24 LHT XT and PR 19 LHT XT [32]). This is not normally observed with similar carbon allotropes (e. g. CNTs) due to their p-doping with the environment [33]. Notably, the S of this grade is lower (in absolute value) than the S of ~ −5 μV∙K^−1^ measured in other Pyrograf^®^ III CNF grades [32]. This clearly reveals that the Pyrograf^®^ III grades (PR 24 LHT XT and PR 19 LHT XT) show higher S-values than the Pyrograf^®^ III PR 25 PS XT used here, which could be related with the lower levels of graphitization of this latter grade.

The functionalized textiles also show negative Seebeck values of ~−1.1 μV∙K^−1^. Therefore, their S is slightly higher (in absolute value) than the S of the as-received CNF powder (−0.60 μV∙K^−1^). This finding confirms the results of the aforementioned work based on dip-coated textiles prepared with Pyrograf^®^ III PR 24 LHT XT [15], since they also presented higher values of S than the as-received CNF powder. Thus, despite its insulating character, the CWF used as a substrate may have some influence on the final S obtained in the coated textiles. In fact, a slight n-doping from the cellulose of CWF to the external graphitic layers of CNFs was recently detected as a possible origin of the S increase with the help of quantum chemical computer models [14]. However, this result could be also related to the effect of the SDBS used in the CNF dispersions. Related to this, some reports have shown that, when SDBS molecules succeed in covering the surface of single-wall carbon nanotubes (SWCNTs), a transfer of electrons from the sodium atoms of SDBS to SWCNTs may occur [34]. Accordingly, the possible existence of SDBS residues on the surface of the CNFs could also justify the higher S values found in the coated textiles.

#### 3.4.3. Power Factor and Figure of Merit

The power factor ($PF=S^{2}\sigma$) of the coated textiles and as-received CNF powder at 30 °C is also shown in Figure 5 and Table 2. The CNF powder presents the highest PF of 4.9 × 10^−5^ μW∙m^−1^ K^−2^, followed by the CWF@6.4 CNF sample with a PF of 2.8 × 10^−5^ μW∙m^−1^ K^−2^. This is worse than the PF of 1.2 × 10^−3^ μW∙m^−1^ K^−2^ found in the samples prepared with 6.4 mg∙mL^−1^ of Pyrograf^®^ III PR 24 LHT XT CNFs [15]. Lastly, the highest figures of merit ($zT=\frac{S^{2}\sigma }{k}T$) of 3.5 × 10^−8^ at 30 °C for CNF powder, followed by 1.96 × 10^−8^ for CWF@6.4 CNF, were calculated from the experimental σ and S obtained here, and the k of 0.43 W∙m^−1^ K^−1^ reported for the paper-like mats of Pyrograf^®^ III [23]. Hence, both values are lower than the zT of 8.7 × 10^−7^ μW∙m^−1^ K^−2^ estimated in coated textiles obtained with inks produced with 6.4 mg∙mL^−1^ of Pyrograf^®^ III PR 24 LHT XT CNFs [15].

### 3.5. Thermoelectric Properties of As-Received CNFs and Dip-Coated Textiles from 30 °C to 100 °C

#### 3.5.1. Electrical Conductivity

The experimental results corresponding to the thermoelectric properties (σ and S) of the as-received CNF powder and functionalized textiles from 30 °C (303.15 K) to 100 °C (373.15 K) are depicted in Figure 6. As shown in Table 2, the CNF powder presents a σ of 136.1 ± 12 S∙m^−1^ at 30 °C, which decreases up to 124.2 ± 7 S∙m^−1^ at 62 °C (335.15 K), and then increases up to 132.4 ± 3 S∙m^−1^ at 94 °C (367.15 K). Thus, over this interval of temperatures (30 °C–94 °C), the CNF powder presents a positive temperature effect (PTC) or dσ/dT < 0, defined as the increase in the electrical resistivity during a heating process. Notably, this matches well to the PTC found for Pyrograf^®^ III CNFs PR 24 LHT XT and PR 19 LHT XT [32]. From this, it can be deduced that, despite the three Pyrograf^®^ III CNFs (PR 24 LHT XT, PR 19 LHT XT and PR 25 PS XT) showing different structures, all of them present similar σ (T) behaviors. However, unlike the CNF powder, the dip-coated textiles show an increase in their conductivity with temperature (dσ/dT > 0) or a negative temperature effect (NTC). For instance, the σ (T) of the CWF@6.4 CNF increases from 23.2 ± 0.7 S∙m^−1^ at 30 °C to 29.5 ± 0.6 S∙m^−1^ at 100 °C (blue symbols in Figure 6a). Interestingly, an NTC effect was also found in the samples prepared with Pyrograf^®^ III CNFs PR 24 LHT XT [15].

#### 3.5.2. Seebeck Coefficient

The black symbols of Figure 6b represent the experimental S (T) of the CNF powder, which shows that its n-type character does not change at all temperatures. In particular, the S of −0.6 μV∙K^−1^ observed at 30 °C increases up to −0.7 μV∙K^−1^ at 94 °C. It is noted that this increase in S (in absolute value) with temperature is also observed in the Pyrograf^®^ III CNFs PR 24 LHT XT and PR 19 LHT XT powders [32] as well as in dip-coated textiles prepared with Pyrograf^®^ III PR 24 LHT XT CNFs [15]. Likewise, the coated textiles show an increase in their experimental Seebeck with a temperature (dS/dT > 0) as seen in Figure 6b. For instance, the S (T) of CWF@6.4 CNF increases from −1.09 μV∙K^−1^ ± 0.03 at 30 °C and −1.30 μV∙K^−1^ ± 0.02 at 65 °C to −1.41 ± 0.01 μV∙K^−1^ at 100 °C (blue symbols in Figure 6b). Interestingly, the S (T) values of the CWF@3.2 CNF and CWF@6.4 CNF samples are quite similar, with −1.09 μV∙K^−1^ ± 0.03 at 30 °C and −1.30 μV∙K^−1^ ± 0.02 at 65 °C, respectively.

### 3.6. Electrical Conductivity σ (T) and Seebeck Coefficient S (T) Modeling of As-Received CNFs and Dip-Coated Textiles from 30 °C to 100 °C

#### 3.6.1. Electrical Conductivity σ (T) Modeling

The σ (T) of all samples is evaluated by the 3D variable range hopping (VRH) model [16]:(1)$$\sigma \left(T\right)=\sigma_{0}\;\exp[\pm {\left(\frac{T_{C}}{T}\right)}^{\frac{1}{4}}]$$

Here, σ_0_ is the electrical conductivity at an infinite temperature, and $T_{C}\equiv \frac{{|W}_{D}|}{k_{B}}$ is a specific temperature scale, where W_D_ is defined as the average energy potential barrier if W_D_ < 0 or as the average potential well if W_D_ > 0, and k_B_ is the Boltzmann´s constant. It is noticed that, when W_D_ > 0 in Equation (1), its dσ/dT > 0 represents a thermally activated hopping mechanism across a random network of potential wells. In contrast, when W_D_ < 0, its dσ/dT < 0 exhibits a thermally activated scattering mechanism across an aleatory distribution of impurities or structural defects. Table 3 presents σ_0_, T_C_, and W_D_ calculated from Equation (1). In particular, values of σ_0_ = 109 S∙m^−1^, T_C_ = 3.8 × 10^−1^ K, and W_D_ = −3.3 × 10^−5^ eV are obtained for the CNFs. Thus, as it happened with the Pyrograf^®^ III CNFs PR 19 LHT XT and PR 24 LHT XT [32], the Pyrograf^®^ III CNF powder used here also shows negative W_D_.

Likewise, the 3D VRH model was used to assess the σ (T) of the functionalized textiles. As Table 3 shows, the T_C_ obtained for them is up to six orders of magnitude higher than the T_C_ of the CNF powder. However, their W_D_ is positive, and up to five orders of magnitude higher than that of the CNF powder. It is noticed that positive W_D_ values were also obtained for dip-coated textiles prepared with Pyrograf^®^ III CNFs PR 24 LHT XT [15]. Hence, the σ (T) of the conductive textiles can be understood as the charge carriers overcoming a random network of potential wells by thermally activated hopping [35], while the σ (T) of the as-received CNF powder is explained as a thermally activated backscattering mechanism [36]. Therefore, it can be concluded from this analysis that the σ (T) observed in the coated textiles cannot be totally interpreted by the σ (T) found in the CNF powder, but the cotton fabric or other factors (such as the remains of the SDBS surfactant) have their relevance to understand their electrical conducting mechanism, as it was commented in the precedent Section 3.4.2.

#### 3.6.2. Seebeck Coefficient S (T) Modeling

The model proposed for describing the nonlinear Seebeck behavior of doped MWCNT mats is used for the S (T) modeling of all samples [17]:(2)$$S\;(T)=bT+\frac{cT_{p}}{T^{2}}\frac{\exp(\frac{T_{P}}{T})}{{\left[\exp(\frac{T_{P}}{T})+1\right]}^{2}}$$

Here, bT represents the metallic (linear) term of S (T), c is a constant, $T_{p}=\left(E_{p}-E_{F}\right)/k_{B}$ where E_F_ is the Fermi energy level, and E_P_ is the energy corresponding to the sharply varying and localized states near E_F_ in the density of states due to the contribution of impurities [36]. All the parameters b, c, T_P_ and $E_{P}-E_{F}$ obtained by Equation (2) are presented in Table 4. The term b of the CNF powder is positive with a value of 1.5 × 10^−1^ μV∙K^−2^, while the T_P_ is 1082.3 K, yielding E_P_ − E_F_ of 9.3 × 10^−2^ eV. Thus, the b, T_P_, and E_P_ − E_F_ found for this Pyrograf^®^ III CNFs PR 25 PS XT are similar to the b, T_P_, and E_P_ − E_F_ are calculated by the same model for Pyrograf^®^ III CNFs PR 19 LHT XT and PR 24 LHT XT [32]. As indicated in Equation (2), there are two mechanisms occurring in parallel. The first one, described by the term bT, represents the contribution from the nearly free charges (metallic contribution). If b > 0, as it is the case for this CNF grade and the dip-coated textiles, then this contribution is due to holes (positive carriers). Interestingly, negative b values, attributed to an n-type doping caused by the textiles or the dip-coating method, were obtained for textiles prepared with Pyrograf^®^ III PR 24 LHT XT CNFs [15], so in this particular point, unlike in precedent work, this grade (PR 25 PS XT) does not produce the same S (T) on their as-derived textiles. The other mechanism described by the second summand of Equation (2) represents the contribution arising from resonances at the density of states near the Fermi level E_F_. Since the term c is negative for the CNF powder and their derivative textiles, this implies that the charge carriers driven by this second mechanism are mainly electrons, which may be caused by impurities or defects present in the CNF structure [17]. It is noticed that the Pyrograf^®^ III CNFs PR 19 LHT XT and PR 24 LHT XT grades also presented c < 0 [32], and the same happened with the electrical conductive textiles prepared with Pyrograf^®^ III PR 24 LHT XT CNFs [15]. As can be seen from Table 4, the fitting values determined for the conductive textiles using Equation (2) are similar to the parameters calculated for the CNF powder, and this analysis concludes that the S (T) of the CNF powder clearly determines the S (T) of their derivative textiles.

## 4. Conclusions

In this work, the electrical conductivity (σ) and Seebeck coefficient (S) between 30 °C and 100 °C of cotton woven fabrics (CWFs) functionalized with aqueous inks made from pyrolytically stripped Pyrograf^®^ III PR 25 PS XT carbon nanofibers (CNFs) via dip-coating method are presented. At 30 °C, the σ, S, and power factor (PF) of the as-received CNFs are ~136 S∙m^−1^, −0.6 μV∙K^−1^, and 5 × 10^−5^ μW∙m^−1^ K^−2^, respectively. The conductive textiles from the inks prepared with the higher amount of CNFs (6.4 mg∙mL^−1^) show lower conductivities of ~23 S∙m^−1^, but higher S (absolute value) of −1.1 μV∙K^−1^ than the as-received CNFs, corresponding to a PF of 2.8 × 10^−5^ μW∙m^−1^ K^−2^ at 30 °C. Moreover, unlike the CNFs, the dip-coated textiles show a dσ/dT > 0 behavior from 30 °C to 100 °C successfully depicted by the 3D variable range hopping (VRH) model, and physically interpreted as a thermally activated hopping of the charge carriers. Moreover, the as-received CNFs and the coated textiles show an enhancement in their S with temperature (dS/dT > 0), properly fitted with the model suggested for some doped multiwall carbon nanotube (MWCNT) mats with nonlinear S (T). Hence, the σ (T) of this pyrolytically stripped Pyrograf^®^ III CNF does not give the complete picture of the σ (T) of their derived textiles, and therefore, the own insulating CWF or other factors (such as the residuals of the surfactant used for the CNF dispersions) may play their contribution in their mechanism conduction. In summary, this study presents the σ (T) and S (T) analyses of n-type conductive textile fabrics easily produced with commercial n-type carbon nanofibers, which could act as potential building blocks of wearable thermoelectric generators (TEGs).

## Figures and Tables

**Figure 1 materials-16-04335-f001:**
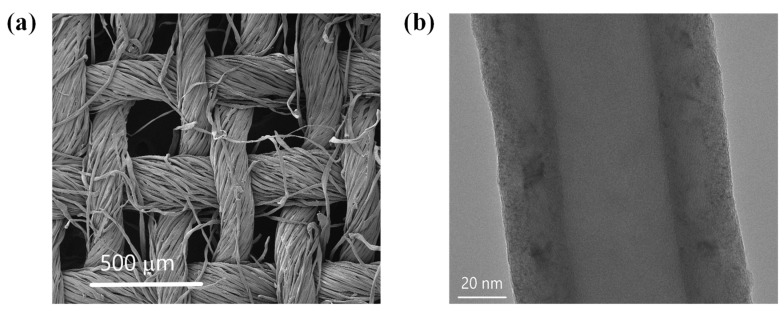
Morphology of the CWF and a single CNF (Pyrograf^®^ III PR 25 PS XT). (**a**) SEM image of the CWF; and (**b**) TEM image of CNF.

**Figure 2 materials-16-04335-f002:**
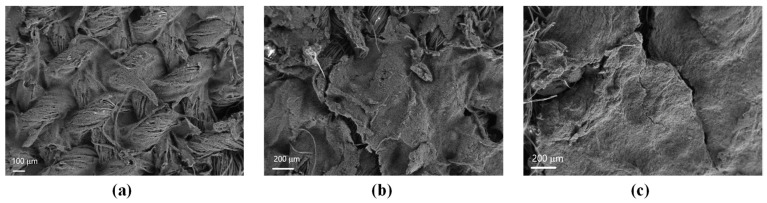
SEM micrographs of coated textiles´ surfaces. (**a**) CWF@1.6 CNF, (**b**) CWF@3.2 CNF, and (**c**) CWF@6.4 CNF.

**Figure 3 materials-16-04335-f003:**
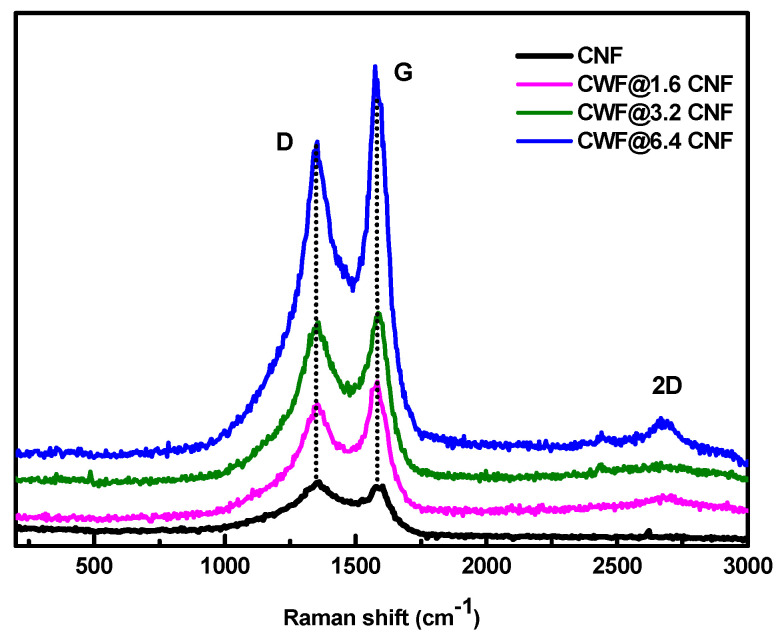
Raman spectra of CNFs and CWF@1.6 CNF, CWF@3.2 CNF. and CWF@6.4 CNF.

**Figure 4 materials-16-04335-f004:**
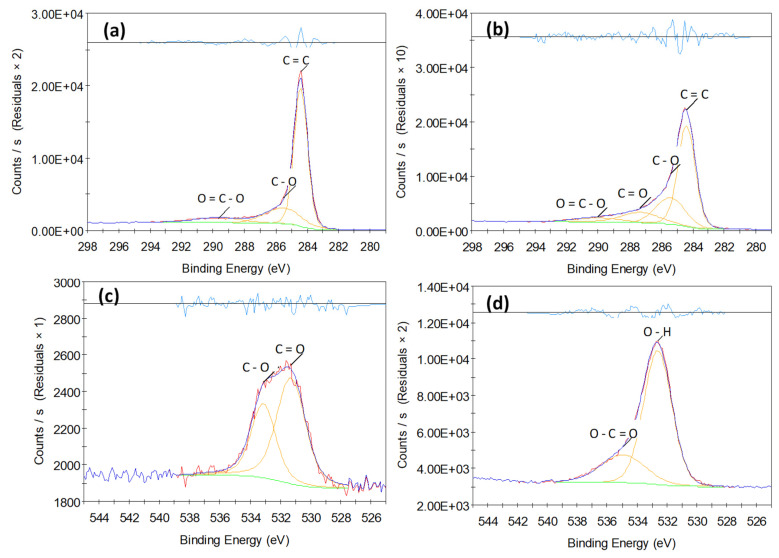
Deconvoluted XPS spectra regarding C1s (**a**,**b**) and O1s (**c**,**d**) core levels and belonging to pristine CNFs (**a**,**c**) and CWF@1.6 CNF (**b**,**d**).

**Figure 5 materials-16-04335-f005:**
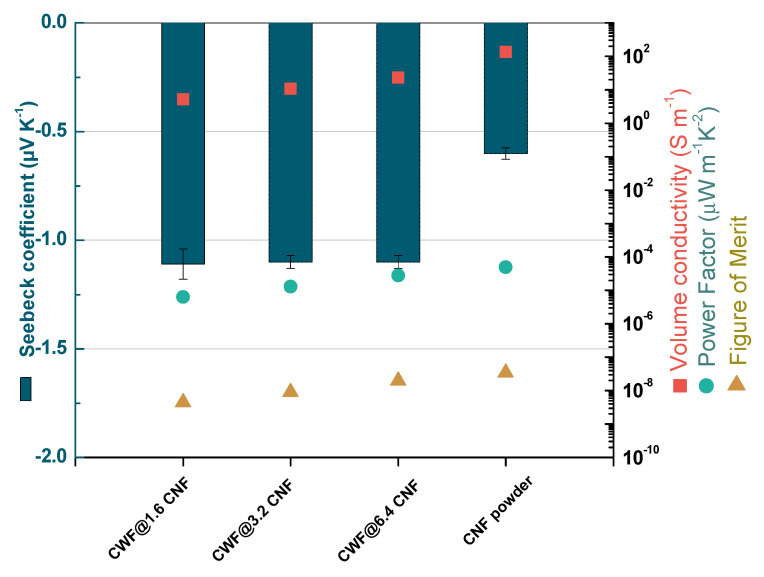
Experimental electrical conductivity (squared symbols), experimental Seebeck coefficient (bars), power factor (circle symbols), and estimated figure of merit (triangle symbols) of conductive textiles and CNFs at 30 °C.

**Figure 6 materials-16-04335-f006:**
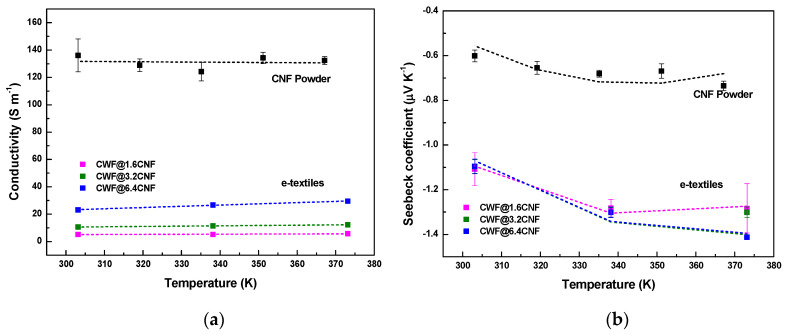
Thermoelectric properties of CNFs and conductive textiles at temperatures from 30 °C to 100 °C. (**a**) Experimental electrical conductivity (squared symbols); and (**b**) experimental Seebeck coefficient (squared symbols). The dashed lines represent the fitting of σ (T) and S (T) with Equations (1) and (2) of Section 3.6, respectively.

**Table 1 materials-16-04335-t001:** Parameters obtained from the fitting of the Raman spectra.

Sample	w_G_ (cm^−1^)	FWHM_G_ (cm^−1^)	w_D_ (cm^−1^)	FWHM_D_ (cm^−1^)	I_D_/I_G_	L_a_ (nm)
CNFs	1587	100	1353	140	1	4.4
CWF@1.6 CNF	1582	100	1350	130	0.81	5.4
CWF@3.2 CNF	1587	100	1352	160	0.89	5
CWF@6.4 CNF	1586	90	1352	140	0.8	5.5

**Table 2 materials-16-04335-t002:** Experimental electrical conductivity (σ), experimental Seebeck (S) coefficient, power factor (PF), and estimated figure of merit (zT) of conductive textiles and CNFs at 30 °C.

Sample	σ (S m^−1^)	S (µV K^−1^)	P F (μW m^−1^ K^−2^)	zT
CWF@1.6CNF	5.14 ± 0.29	−1.11 ± 0.07	6.33 × 10^−6^	4.45 × 10^−9^
CWF@3.2CNF	10.64 ± 0.43	−1.10 ± 0.03	1.29 × 10^−5^	9.0 × 10^−9^
CWF@6.4CNF	23.19 ± 0.68	−1.10 ± 0.03	2.81 × 10^−5^	1.96 × 10^−8^
CNF powder	136.09 ± 12	−0.60 ± 0.03	4.92 × 10^−5^	3.47 × 10^−8^

**Table 3 materials-16-04335-t003:** Parameters σ_0_, T_C_, and W_D_ obtained by fitting the experimental values of σ (T) with the VRH model Equation (1).

Sample	σ_0_ (S m^−1^)	T_C_ (K)	W_D_ (eV)
CWF@1.6 CNF	33.4	3.8 × 10^3^	3.3 × 10^−1^
CWF@3.2 CNF	175.4	1.9 × 10^4^	1.6
CWF@6.4 CNF	2694.5	1.5 × 10^5^	13.3
CNF Powder	109.1	3.8 × 10^−1^	−3.3 × 10^−5^

**Table 4 materials-16-04335-t004:** Parameters b, c, T_P_, and $E_{P}-E_{F}$ obtained by fitting the experimental values of S (T) with Equation (2).

Sample	b (μVK^−2^)	c (μV)	T_p_ (K)	E_p_ − E_F_ (eV)
CWF@1.6 CNF	1.4 × 10^−2^	−1.7 × 10^4^	1073.7	9.2 × 10^−2^
CWF@3.2 CNF	1.2 × 10^−2^	−1.6 × 10^4^	1109.2	9.6 × 10^−2^
CWF@6.4 CNF	1.2 × 10^−2^	−1.6 × 10^4^	1107.6	9.5 × 10^−2^
CNF Powder	1.5 × 10^−2^	−1.6 × 10^4^	1082.3	9.3 × 10^−2^

## Data Availability

The data presented in this study are available upon request from the corresponding author after obtaining permission from an authorized person.

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
