# Peer review of "Thermoelectric Properties of Cotton Fabrics Dip-Coated in Pyrolytically Stripped Pyrograf® III Carbon Nanofiber Based Aqueous Inks"

_materials, 2023, doi:10.3390/ma16124335_

Round 1
Reviewer 1 Report
1. Please improve the introduction section. Only 12 references are cited in the introduction section.
2. It is not recommended to add te figures to the materials section. Please move the SEM images to Figure 2.
3. Figure 3. Have the authors deconvoluted the Raman spectra. It is required as a quatitative analysis being performed. Please show the deconvolution.
4. Figure 4 - Have the authors plotted the XPS data using a graphing software. The resolution of the figures need to be improved.
5. Try to do a quatitaive analysis using XPS as well.
Up to the standard
Reviewer 2 Report
The article is devoted to a relevant topic, but it would be good to make some corrections to accept it for publication.
1. Finish the Introduction section with the overall goal of the work.
2. Perhaps the Methodology section should start with a general plan for the whole work. Indicate which experiments were performed, and how many samples were used, and in sections 2.1 and 2.2, describe each method in detail.
3. A similar proposal applies to section 3. At the beginning of the section, give a general description of what you are doing so that the reader can immediately understand what work is being done with the materials under study, and then already give the results of various tests in sections 3.1, 3.2, ...
4. You describe and analyze separately each type of fabric test. You need to move the general description into section 3.6. In this section, link the results obtained into a single picture and make a comprehensive discussion of them.
5. In the conclusions or section 3.6, describe the possibilities of the practical application of the results obtained.
Round 2
Reviewer 1 Report
The authors have addressed the comments suggested by the reviewer.
Author Response
We appreciate very much the time and interest in our work of Reviewer.
Reviewer 2 Report
In general, the article was finalized by the authors.
It would be good to give the answer to the second question at the beginning of the Methodology section.
Author Response
We appreciate very much the time and interest in our work of the Reviewer, and we understand that the response to his/her second question has been responded in our previous report.